# Vision-aligned Latent Reasoning for Multi-modal Large Language Model

**Byungwoo Jeon** [1]  **Yoonwoo Jeong** [2]  **Hyunseok Lee** [1]  **Minsu Cho** [2 3 †]  **Jinwoo Shin** [1 3 †]

## Abstract

Despite recent advancements in Multi-modal Large Language Models (MLLMs) on diverse understanding tasks, these models struggle to solve problems which require extensive multi-step reasoning. This is primarily due to the progressive dilution of visual information during long-context generation, which hinders their ability to fully exploit test-time scaling. To address this issue, we introduce **Vision-aligned Latent Reasoning (VaLR)**, a simple, yet effective reasoning framework that dynamically generates vision-aligned latent tokens before each Chain-of-Thought reasoning step, guiding the model to reason based on perceptual cues in the latent space. Specifically, VaLR is trained to preserve visual knowledge during reasoning by aligning intermediate embeddings of MLLM with those from vision encoders. Empirical results demonstrate that VaLR consistently outperforms existing approaches across a wide range of benchmarks requiring long-context understanding or precise visual perception, while exhibiting test-time scaling behavior not observed in prior MLLMs. In particular, VaLR improves the performance significantly from 33.0% to 52.9% on VSI-Bench, achieving a 19.9%p gain over Qwen2.5-VL. Code is available at project page.

## 1. Introduction

Multi-modal Large Language Models (MLLMs) have achieved remarkable success in various multi-modal tasks such as image captioning (Cheng et al., 2025) and visual question answering (Manmadhan & Kovoor, 2020; Huynh et al., 2025). Beyond these tasks, there is a growing demand to deploy MLLMs in more complex applications that require multi-step reasoning and long-horizon planning, such

as computer-use agents (CUA) (Anthropic, 2024a;b) and Vision-Language-Action (VLA) models (Kim et al., 2024; Black et al., 2024; Lee et al., 2025; Bjorck et al., 2025). A core challenge in such applications lies in integrating perceptual information into multi-step logical reasoning within MLLM architectures.

In the language domain, Chain-of-Thought (CoT) (Wei et al., 2022) has emerged as a cornerstone for improving reasoning capabilities of LLMs, enabling LLMs to decompose intricate tasks into intermediate linguistic steps. Building on the success of CoT, recent studies (Zheng et al., 2026; Li et al., 2025b) have extended this approach from LLMs to MLLMs. However, in contrast to the test-time scaling law (Snell et al., 2024) of LLMs, MLLMs frequently struggle with long-context reasoning due to the attenuation of visual signals as the generated sequence length increases.

To address this issue, recent research in MLLMs focuses on enhancing the long-context reasoning of MLLMs. For instance, a line of work strengthens text reasoning of MLLMs via supervised fine-tuning (Yue et al., 2024; Yu et al., 2024) or reinforcement learning (Wang et al., 2024b; Havrilla et al., 2024; Shao et al., 2024b; Yu et al., 2025a). While these text-centric methods have shown significant progress, they still suffer from diminishing visual signals when generating long text sequences. Alternatively, several studies explicitly re-introduce visual information by interleaving visual tokens (Zheng et al., 2026; Yang et al., 2026; Yoon et al., 2025) or generating images (Wang et al., 2025a; Li et al., 2025b). Yet, these approaches rely on static single-view visual features and use them only as a fixed initial context. Throughout this work, we demonstrate that utilizing static visual features leads to the gradual loss of visual context, whereas dynamically allocating visual details at each reasoning stage ensures information preservation, thereby enabling robust long-context reasoning in MLLMs.

In this paper, we introduce Vision-aligned Latent Reasoning (VaLR), a novel multi-modal reasoning framework that generates vision-aligned latent tokens during the reasoning process, which is inspired by the latent reasoning LLM approach (Hao et al., 2024b). The core idea of VaLR is to inject learnable latent tokens before each text-based reasoning step, creating "visual checkpoints" that keep the reasoning process grounded in image details. Unlike standard text

---

†Equal advising. [1]KAIST [2]POSTECH [3]RLWRLD. Correspondence to: Jinwoo Shin <jinwoos@kaist.ac.kr>.

*Proceedings of the 43$^{rd}$ International Conference on Machine Learning*, Seoul, South Korea. PMLR 306, 2026. Copyright 2026 by the author(s).

tokens, these latent tokens are explicitly supervised to learn consistency with the dense visual representations of the input image which is highly correlated with the subsequent reasoning step. Specifically, we introduce a two-stage curriculum learning framework to gradually equip MLLMs with latent reasoning capabilities. The first stage involves supervised fine-tuning on general vision question-answering datasets to learn fundamental multi-modal reasoning ability. In the second stage, we incorporate a new group of latent tokens before every CoT step. We then apply representation alignment (Yu et al., 2025b) to these latent tokens with dense features extracted from the corresponding image frame by vision encoders, *e.g.*, DINOv2/v3 (Oquab et al., 2023; Siméoni et al., 2025), CLIP (Radford et al., 2021), and SigLIPv2 (Tschannen et al., 2025).

We demonstrate the effectiveness of VaLR through extensive evaluations on multiple Vision Question-Answering (VQA) datasets. Overall, VaLR exhibits superior performance over existing baselines on multiple VQA benchmarks. Specifically, on VSI-Bench (Yang et al., 2025b), VaLR boosts the accuracy of Qwen2.5-VL from 33.0% to 52.9%. Notably, as shown in Figure 2, VaLR successfully follows the test-time scaling law: the performance of VaLR improves in cases requiring longer reasoning, whereas baselines degrade under similar conditions. Furthermore, ablation studies suggest that VaLR can be used agnostically on several vision encoders, *e.g.*, DINO, SigLIP, CLIP and even works with the standalone vision encoders of the original MLLM, *i.e.*, Qwen2.5-VL encoder (Bai et al., 2025).

## 2. Related Works

**Multi-modal Large Language Models (MLLMs).** Recent advancements in MLLMs harness the inherent reasoning proficiency of LLMs to establish unified architectures designed to handle multiple modalities within a single framework. Pioneering studies integrate visual information into LLMs, predominantly utilizing either resamplers (Alayrac et al., 2022; Awadalla et al., 2023; Li et al., 2025a; Cha et al., 2024) or Q-Former (Li et al., 2023; Dai et al., 2023; Zhu et al., 2023; Lin et al., 2024). Despite the effectiveness of these specialized architectures, LLaVA (Liu et al., 2023; 2024a) and its successors (Chen et al., 2024a; Liu et al., 2024b; Chu et al., 2023; 2024; Bai et al., 2023a;b; Yang et al., 2024; 2025a) demonstrate that aligning each modality through a trainable lightweight projector is sufficient when paired with visual instruction tuning. Nevertheless, these models suffer from solving problems that require comprehensive reasoning, falling short of the reasoning capabilities exhibited by Chain-of-Thought (CoT).

**Chain-of-Thought (CoT) and Latent Reasoning.** The emergence of Chain-of-Thought (CoT) prompting has significantly enhanced the reasoning capabilities of large lan-

guage models (LLMs) by decomposing complex problems into intermediate linguistic steps. While early works (Wei et al., 2022; Khot et al., 2023; Zhou et al., 2023) primarily relied on explicit prompting to derive these chains, subsequent research has focused on intrinsic enhancement through supervised fine-tuning (Yue et al., 2024; Yu et al., 2024) or reinforcement learning (Wang et al., 2024b; Havrilla et al., 2024; Shao et al., 2024b; Yu et al., 2025a). To expand the search space of reasoning chains during inference, extensive studies have introduced tree-based (Xie et al., 2023; Yao et al., 2023; Hao et al., 2024a) and trajectory-based (Lehnert et al., 2024; Gandhi et al., 2024; Su et al., 2025) exploring algorithms. Motivated by the insight that natural language cannot encapsulate all forms of reasoning, recent paradigms have shifted toward operating directly within latent space (Hao et al., 2024b; Wang et al., 2026a; Li et al., 2026b) or learning to generate visual information from latent reasoning features (He et al., 2024; Li et al., 2025b). Orthogonally, several approaches (Zheng et al., 2026; Yang et al., 2026) propose to interleave visual tokens in reasoning trajectories to empower multi-modal reasoning. In this work, we align the latent reasoning tokens with features from vision encoders to facilitate visual reasoning within the latent space.

**Leveraging External Vision Encoders in MLLMs.** Recent MLLMs incorporate rich visual features, *e.g.*, CLIP (Radford et al., 2021), DINO (Oquab et al., 2023; Siméoni et al., 2025), SigLIP (Zhai et al., 2023), and VGGT (Wang et al., 2025b), to enhance their visual and spatial reasoning capabilities. For instance, PrismaticVLM (Karamcheti et al., 2024) integrates CLIP and DINO features through trainable projection layers to leverage rich visual representations. Similarly, PaliGemma (Beyer et al., 2024) exploits the dense features of SigLIP to enable comprehensive visual understanding with fewer parameters. To enhance the spatial awareness of MLLMs, several studies (Zheng et al., 2025; Wu et al., 2025; Huang et al., 2025) leverage VGGT to inject token-wise spatial information. Concurrent with our work, CoVT (Qin et al., 2025) and Monet (Wang et al., 2026a) enhance the visual understanding of MLLMs by leveraging rich visual features to perform reasoning directly within the visual space. However, as the reasoning chain lengthens, they suffer from diminishing visual signals since the visual information is only utilized as a fixed initial context. To alleviate this attenuation, our VaLR aligns latent tokens at each reasoning step with vision encoders, thereby enabling long-context visual reasoning.

## 3. Vision-aligned Latent Reasoning

We propose VaLR, an approach that aligns latent reasoning tokens with visual features to prevent visual signal decay, thereby enabling effective test-time scaling in MLLMs.

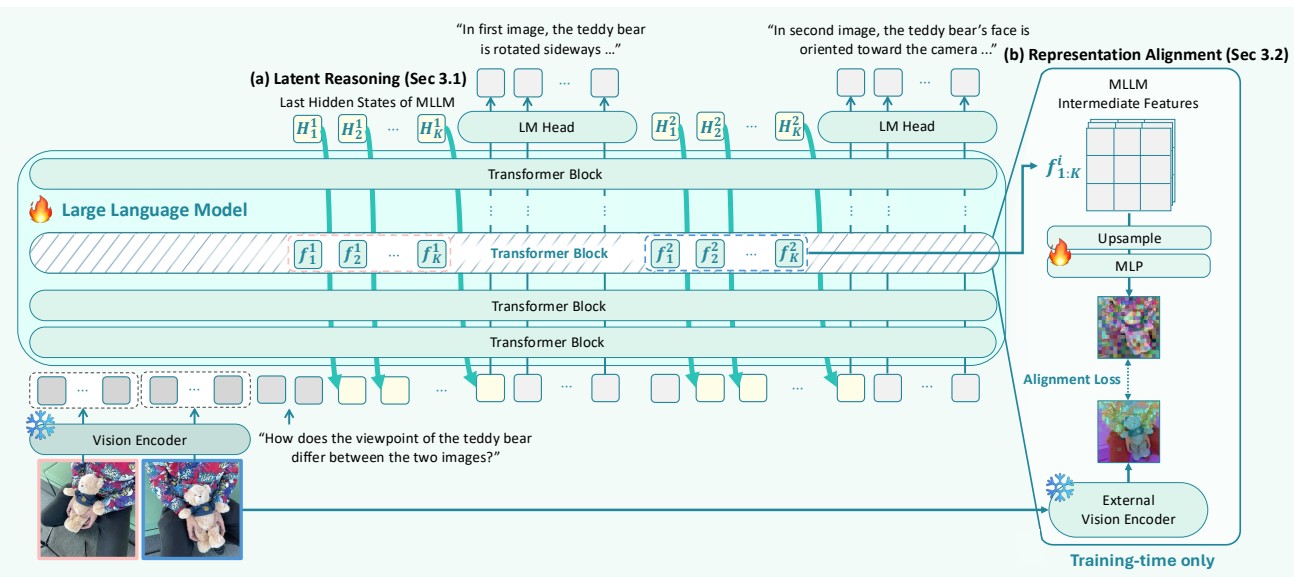

*Figure 1.* **Overview of VaLR.** Our framework, VaLR, generates vision-aligned latent tokens and language tokens throughout reasoning process. (a) During latent token generation, the last hidden states of MLLM becomes input embedding for the next token prediction. (b) To train the latent token generation, we align the intermediate features of MLLM with pre-trained visual representation extracted from external vision encoders. Note that we do not use the external vision encoder at test-time.

In Section 3.1, we first revisit the concept of latent reasoning in MLLMs. Then, in Section 3.2, we discuss how multi-modal reasoning can be enhanced through representation alignment between MLLMs and vision encoders. Finally, Section 3.3 presents VaLR, a two-stage supervised fine-tuning (SFT) pipeline designed to gradually equip MLLMs with latent multi-modal reasoning capabilities. The overall pipeline of VaLR is illustrated in Figure 1.

### 3.1. Latent Reasoning in MLLMs

Formally, given an input text sequence $x = (x_1, \ldots, x_T)$ and images $\mathcal{I}$, we formulate the task as generating a corresponding text response. During inference with latent reasoning, the model iteratively switches between two distinct modes: latent and language. In detail, in the latent mode, the model produces latent reasoning tokens that are not directly shown as text, while in the language mode, it generates the response with text tokens.

Specifically, the native vision encoder first extracts image tokens from images $\mathcal{I}$, *i.e.*, $v = (v_1, v_2, \cdots v_S) = \text{ViT}(\mathcal{I})$. Subsequently, the transformer decoder processes input text-token embeddings, $E_T = [v_1, \cdots, v_S, e(x_1), ..., e(x_T)]$, to yield the last hidden state $H_T = \text{Transformer}(E_T)$, where $e$ is the token embedding function. During inference, the model enters the latent mode by predicting a special token $<\text{latent}>$ and reverts to the language mode by predicting another special token $<\backslash\text{latent}>$. In the latent mode, the model leverages the previous hidden state, $h_t = H_t[t, :]$, as input for the next prediction, whereas in the language mode, the model uses the token embedding, $e(x_{t+1})$, as input for

the next prediction, as formulated below:

$$E_{t+1} = \begin{cases} [E_t; h_t] & \text{if latent mode,} \\ [E_t; e(x_{t+1})] & \text{if language mode,} \end{cases}$$
$$H_{t+1} = \text{Transformer}(E_{t+1}),$$

where $t > T$. This recursive process repeats until the model predicts the $<\text{EOS}>$ token. Upon entering the latent mode, the model is constrained to remain in this state for a fixed number $K$ of steps. After $K$ latent steps, the model reverts to the language mode and resumes generating text tokens from the current hidden state $h_t$, using the language model head, LM-Head:

$$\mathcal{M}(x_t | v, x_{<t}) = \text{LM-Head}(h_t),$$

where $\mathcal{M}$ denotes the standard MLLM. This alternation strategy allows MLLMs to broaden its reasoning capability without explicit linguistic reasoning steps.

### 3.2. Latent Reasoning with Representation Alignment

To effectively leverage latent reasoning for visual grounding, we align hidden states of MLLM with visual features from pre-trained vision encoders during the latent mode. This alignment encourages the MLLM to maintain visual information throughout the recurrent reasoning process.

**Alignment objective.** For each reasoning stage $i$, we first select an image $I^{(i)} \in \mathcal{I}$ (details in Appendix B). We then extract patch-wise visual features from pre-trained vision encoder, $\phi$, *i.e.*, $\mathbf{F}_\phi^{(i)} = \phi(I^{(i)}) \in \mathbb{R}^{P \times D}$, where $P$ is the

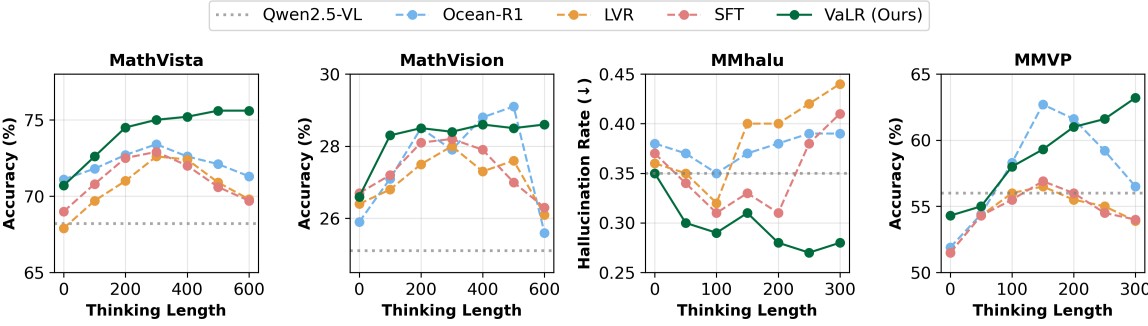

*Figure 2.* **Reasoning length-wise analysis.** We investigate the effect of reasoning length on model performance across different MLLMs. We report hallucination rate on MMhalu (Sun et al., 2024) benchmark and accuracy (%) on MathVista (Lu et al., 2024), MathVision (Wang et al., 2024a), and MMVP (Tong et al., 2024b) benchmarks. For MMhalu, lower is better. We observe that VaLR is the only method that exhibits consistent performance improvements as reasoning length increases, while remaining robust on long-horizon tasks.

number of patches and $D$ is the feature dimension. Afterward, we extract features from the intermediate layer of MLLM, *i.e.*, $\mathbf{F}_{\mathrm{MLLM}}^{(i)} = [f_1^{(i)}, \cdots, f_K^{(i)}]$. We project these intermediate features through a learnable MLP $\psi$ to match the dimension of vision encoder features:

$$\hat{\mathbf{F}}_{\mathrm{MLLM}}^{(i)} = \psi \left( \mathrm{Upsample} \left( \mathbf{F}_{\mathrm{MLLM}}^{(i)} \right) \right) \in \mathbb{R}^{P \times D},$$

where the 'Upsample' denotes an operation that aligns the image feature resolution of the MLLM with that of the pre-trained vision encoder. The representation alignment loss, *i.e.*, $\mathcal{L}_{\mathrm{REPA}}$, encourages these projected latent features to align with the visual features using patch-wise cosine similarity throughout all latent reasoning stages:

$$\mathcal{L}_{\mathrm{REPA}} := -\frac{1}{NP} \sum_{i=1}^{N} \sum_{p=1}^{P} \mathrm{sim} \left( \hat{\mathbf{F}}_{\mathrm{MLLM}}^{(i)}[p, :], \mathbf{F}_{\phi}^{(i)}[p, :] \right),$$

where $\mathrm{sim}(\cdot, \cdot)$ denotes the conventional cosine similarity function. By aligning with visual features, each latent token learns to encode visual information inherent in the image, thereby enabling comprehensive visual reasoning. Note that the alignment is applied only during training, while at inference time the model performs latent mode reasoning without REPA supervision, relying on learned visual grounding.

**Multi-encoder Alignment.** While alignment with a single vision encoder provides a robust visual foundation, we observe that leveraging multiple vision encoders enables the model to capture complementary visual representations. For instance, CLIP (Radford et al., 2021) and SigLIP (Tschannen et al., 2025) excel at semantic understanding, DINO (Oquab et al., 2023; Siméoni et al., 2025) captures fine-grained appearance and spatial relationships, and $\pi^3$ (Wang et al., 2026b) encodes 3D spatial structure. To leverage these complementary strengths, we extend our framework to incorporate multiple vision encoders simultaneously.

Let $\{\phi_1, \cdots, \phi_M\}$ denotes a set of $M$ frozen vision encoders. We extract features from each vision encoder for

each reasoning stage $i$:

$$\mathbf{F}_{\phi_m}^{(i)} = \phi_m(I^{(i)}) \in \mathbb{R}^{P_m \times D_m} \quad \text{for } m = 1, \cdots, M,$$

where $P_m$ and $D_m$ denote the varying number of patches and feature dimension across different vision encoders, respectively. For each vision encoder, we employ a separate learnable projection head $\psi_m$ to match its feature dimension. The multi-encoder alignment loss is computed as the average of individual REPA losses:

$$\mathcal{L}_{\mathrm{REPA}}^{\mathrm{multi}} := \frac{1}{M} \sum_{m=1}^{M} \mathcal{L}_{\mathrm{REPA}}^{(m)},$$

where each $\mathcal{L}_{\mathrm{REPA}}^{(m)}$ follows the same formulation as the single-encoder case but uses features from the $m$-th vision encoder, $\phi_m$, and its corresponding projection head $\psi_m$. This multi-encoder approach allows the model to distill diverse visual knowledge into its latent reasoning space, enhancing both spatial awareness and general visual understanding.

### 3.3. Training Pipeline

We adopt a two-stage curriculum learning strategy to progressively foster latent reasoning in MLLMs. In the first stage, we perform standard supervised fine-tuning (SFT) on Chain-of-Thought (CoT) visual question-answering (VQA) datasets to establish foundational multi-modal reasoning capabilities. Subsequently, in the second stage, we decompose the reasoning into step-by-step phases and interleave latent reasoning tokens, allowing the model to reason within the latent representations. Crucially, we employ representation alignment (REPA) to align the intermediate hidden states of the MLLM with features extracted from vision encoders such as DINO (Oquab et al., 2023; Siméoni et al., 2025), CLIP (Radford et al., 2021), or SigLIP (Tschannen et al., 2025). This alignment empowers MLLMs to retain visual information required for reasoning, thereby enabling robust long-context reasoning.

*Table 1.* **Main results on long-context evaluation.** Accuracy (%) on multi-view VQA Benchmark, VSI-Bench (Yang et al., 2025b), for VaLR (Ours) and other baselines including several reasoning models, the base model, and latent reasoning models. We report 8 different sub-task accuracy and average (Avg.) accuracy. VaLR-S and VaLR-M denote single encoder (DINOv3)-aligned model and multiple encoder (DINOv3, SigLIPv2, $\pi^3$)-aligned model, respectively. The bold indicates the best result and underlined indicates the second best result within the group.

| Method | Avg. | Obj. Cnt. | Abs. Dist. | Obj. Size | Room size | Rel. Dist. | Rel. Dir. | Route plan | Appr. Order |
|---|---|---|---|---|---|---|---|---|---|
| *Other Models* | | | | | | | | | |
| GPT-4o | 34.0 | 46.2 | 5.3 | 43.8 | 38.2 | 37.0 | 41.3 | 31.5 | 28.5 |
| LLaVA-NeXT-Video-7B | 35.6 | 48.5 | 14.0 | 47.8 | 24.2 | 43.5 | 42.4 | 34.0 | 30.6 |
| *Reasoning Models* | | | | | | | | | |
| R1-OneVision-7B (Yang et al., 2025c) | 16.1 | 15.0 | 1.7 | 0.5 | 2.8 | 26.5 | 40.0 | 24.2 | 14.7 |
| Ocean-R1-7B (Lingfeng et al., 2025) | 30.5 | 16.5 | 14.6 | 38.9 | 40.1 | 38.0 | 36.1 | 30.9 | 30.1 |
| *Base Model* | | | | | | | | | |
| Qwen2.5-VL-7B (Bai et al., 2025) | 33.0 | 40.9 | 14.8 | 43.4 | 20.7 | 38.6 | 38.5 | 33.0 | 29.8 |
| + vanilla SFT | 33.7 | 42.3 | 14.7 | 44.1 | 20.8 | 39.4 | 34.7 | 32.5 | 33.5 |
| *Latent Reasoning Models* | | | | | | | | | |
| + LVR (Li et al., 2026b) | 18.4 | 21.4 | 3.6 | 1.4 | 9.0 | 35.1 | 30.9 | 32.0 | 23.1 |
| + CoVT (Qin et al., 2025) | 18.6 | 16.5 | 2.3 | 1.0 | 7.3 | 35.9 | 33.0 | 25.8 | 30.4 |
| + Monet (Wang et al., 2026a) | 14.0 | 1.9 | 0.1 | 0.0 | 0.0 | 38.0 | 20.5 | 24.2 | 31.2 |
| **+ VaLR-S (Ours)** | 41.5 | 49.0 | 24.5 | 53.9 | 38.2 | 43.9 | 41.9 | 34.0 | 39.2 |
| **+ VaLR-M (Ours)** | **52.9** | **66.4** | **40.6** | **64.2** | **56.6** | **50.0** | **51.8** | **35.1** | **48.9** |

**Stage 1: Standard SFT on CoT datasets.** We perform standard SFT on pre-trained MLLMs using 450K samples from existing CoT datasets, endowing MLLMs with language-based reasoning capabilities. Concretely, given a training sample with an input image set $\mathcal{I}$, a question $q$, and ground-truth language CoT reasoning $\mathbf{y} = [r^1, r^2, \cdots, r^N, a]$ where $r^i$ represents the $i$-th reasoning step and $a$ is the final answer, we optimize the model using the standard autoregressive language modeling objective:

$$\mathcal{L}_{\text{CE}} := -\mathbb{E}_{(\mathcal{I},q,y)} \left[ \sum_t \log \mathcal{M}(y_t | v, q, y_{<t}) \right],$$

where $y_t$ denotes the $t$-th token in the reasoning sequence. This stage establishes the fundamental ability to decompose complex visual questions into intermediate linguistic reasoning steps. During this stage, we only train the decoder of MLLM while freezing the native vision encoder.

**Stage 2: Latent token training with REPA.** Building on the standard CoT reasoning capabilities established in Stage 1, we introduce latent reasoning supervised by vision encoders in this stage. We first tailor existing CoT datasets for latent reasoning and then train the model on the tailored datasets using representation alignment (REPA) (Yu et al., 2025b).

Specifically, each sample from existing CoT datasets consists of visual information $v$, a question $q$ conditioned on visual input, a sequence of intermediate reasoning steps $\{r^{(i)}\}_{i=1}^N$, where $N$ denotes the number of reasoning steps, and the corresponding answer $a$, *i.e.*,:

$$v, q \rightarrow \left( r^{(i)} \right)_{i=1}^N \rightarrow a.$$

To adapt these datasets for latent reasoning, we insert $K$

latent tokens, $\{\ell_k^{(i)}\}_{k=1}^K$, before each language reasoning step $r^{(i)}$. To inform the model when the latent mode should be initialized or terminated, we set the first and last tokens of each latent segment to special control tokens, *i.e.*, $\ell_1^{(i)} = $ <latent> and $\ell_K^{(i)} = $ </latent>. This transformation yields a latent-augmented reasoning sequence, which can be expressed as follows:

$$v, q \rightarrow \left( \ell_{[1:K]}^{(i)}, r^{(i)} \right)_{i=1}^N \rightarrow a.$$

In this stage, we extend the Stage 1 training objective with a REPA loss, *i.e.*, $\mathcal{L} := \mathcal{L}_{\text{CE}} + \lambda \mathcal{L}_{\text{REPA}}$. When we use multiple encoders for training, we apply the multi-REPA loss instead of the single-REPA loss, *i.e.*, $\mathcal{L} := \mathcal{L}_{\text{CE}} + \lambda \mathcal{L}_{\text{REPA}}^{\text{multi}}$. We freeze the vision encoder and train only the MLLM decoder. Remark that the REPA loss ensures that the hidden states remain grounded in visual information.

## 4. Experiment

We provide an empirical evaluation of Vision-aligned Latent Reasoning (VaLR) by investigating following questions:

- Does VaLR improve the performance on VQA datasets? (Table 1, Table 2)

- Does VaLR retain performance during long-context reasoning? (Figure 2)

- Does the latent token component really contribute to long-context reasoning? (Table 3)

- Can VaLR be adapted to various vision models and tasks in a model-agnostic manner? (Table 4, Table 5)

*Table 2.* **Main results on perception evaluation.** Accuracy (%) for VaLR and other baselines. including several reasoning models, base model and latent reasoning models. We consider perception evaluation benchmarks, including BLINK, MMVP, MMStar, V*, and CVBench. VaLR-S and VaLR-M denote single encoder (DINOv3)-aligned model and multiple encoder (DINOv3, SigLIPv2, $\pi^3$)-aligned model, respectively. The bold indicates the best result and underlined indicates the second best result within the group.

| Method | BLINK | MMVP | MMStar | V* | CVBench |
|---|---|---|---|---|---|
| *API models* | | | | | |
| GPT-4o | 63.0 | **68.7** | 65.2 | 42.9 | 79.2 |
| Claude-4-Sonnet | 39.6 | 48.7 | 58.8 | 15.2 | 76.3 |
| *Reasoning models* | | | | | |
| R1-OneVision-7B (Yang et al.) | 50.1 | 48.7 | 55.6 | 59.2 | 67.2 |
| Ocean-R1-7B (Lingfeng et al.) | 56.8 | 58.0 | 62.6 | 78.0 | 78.1 |
| *Base model* | | | | | |
| Qwen2.5-VL-7B (Bai et al.) | 55.7 | 56.0 | 67.1 | 76.4 | 74.5 |
| + vanilla SFT | 56.6 | 58.7 | 67.5 | 78.0 | 77.0 |
| *Latent Reasoning Models* | | | | | |
| + LVR (Li et al.) | 52.8 | 59.3 | 64.4 | 81.7 | 76.9 |
| + CoVT (Qin et al.) | 56.0 | 58.7 | 69.2 | 78.0 | 80.0 |
| + Monet (Wang et al.) | 49.1 | 50.0 | 53.3 | 83.3 | 71.1 |
| + VaLR-S (Ours) | 63.1 | 60.3 | 70.8 | 86.4 | 83.1 |
| + VaLR-M (Ours) | **64.7** | 60.3 | **72.3** | **86.9** | **87.6** |

## 4.1. Experimental Setup

**Training Setup.** For the main experiment, we trained VaLR on Qwen2.5-VL-7B (Bai et al., 2025). Unless mentioned otherwise, we use DINOv3 (Siméoni et al., 2025) as the aligning vision encoder. For the analysis and multi-encoder alignment training setting, we additionally consider alternative vision encoders, *e.g.*, DINOv2 (Oquab et al., 2023), CLIP (Radford et al., 2021), SigLIPv2 (Tschannen et al., 2025)), and $\pi^3$ (Wang et al., 2026b). We perform training on 450K scale of Chain-of-Thought (CoT) dataset for both training stages. We construct the dataset with the mixture of several open-source datasets, *e.g.*, Zebra-CoT (Li et al., 2026a), CogCoM (Qi et al., 2025), ReFocus (Fu et al., 2025), Visual-CoT (Shao et al., 2024a), OneThinker-SFT (Feng et al., 2026), and GCoT (Chen et al., 2026). Further details are provided in Appendix A.1.

**Evaluation Setup.** Following the evaluation setup of previous benchmarks, we mainly report the accuracy (%) across all benchmarks. For response generation, we apply greedy sampling. The models are evaluated on various VQA benchmarks, including VSI-Bench, BLINK, MMVP, MMStar, MathVision, and more. In Appendix A.2, we provide additional results under specific evaluation settings.

**Baselines.** We compare VaLR with API, reasoning, supervised finetuned, and latent reasoning models in MLLMs, namely, GPT-4o, Claude-4-Sonnet, R1-OneVision-7B (Yang et al., 2025c), Ocean-R1-7B (Lingfeng et al., 2025), LVR (Li et al., 2026b), CoVT (Qin et al., 2025), and Monet (Wang et al., 2026a).

*Table 3.* **Effect of representation alignment component.** We ablate the latent alignment training component of VaLR. We compare the VaLR without visual alignment (VA) and visual alignment with Qwen encoder (QE) and DINOv3 (Ours). We report accuracy (%) on VSI-Bench, BLINK, MMVP, V*, and CVBench. The bold indicates the best result within the group.

| Method | VSI-Bench | BLINK | MMVP | V* | CVBench |
|---|---|---|---|---|---|
| Qwen2.5-VL-7B | 33.0 | 55.7 | 56.0 | 76.4 | 74.5 |
| + vanilla SFT | 33.7 | 56.6 | 58.7 | 78.0 | 77.0 |
| + VaLR w/o VA | 34.0 | 57.1 | 56.7 | 75.9 | 73.4 |
| + VaLR w/ QE | 39.6 | 58.9 | 60.0 | 81.7 | 81.6 |
| **+ VaLR (Ours)** | **41.5** | **63.1** | **60.3** | **86.4** | **83.1** |

## 4.2. 3D Spatial Reasoning Tasks

We examine VaLR's effectiveness in long-context reasoning by comparing performance on 3D multi-view benchmark, VSI-Bench (Yang et al., 2025b), which requires long-context reasoning ability to integrate spatial information across multiple viewpoints. We report accuracy on 8 sub-tasks and the average accuracy. We train the VaLR with two different setups: (i) VaLR-S, which aligns a single encoder using DINOv3 (Siméoni et al., 2025) and (ii) VaLR-M, which aligns multiple encoders using DINOv3, SigLIPv2 (Tschannen et al., 2025), and $\pi^3$ (Wang et al., 2026b).

**Results.** As shown in Table 1, VaLR-S achieves an average accuracy of 41.5%, substantially outperforming the base model, Qwen2.5-VL (33.0%). In contrast, previous latent reasoning methods struggle on this benchmark requiring multi-view understanding. For example, Monet (Wang et al., 2026a) reaches 14% in average accuracy, and other models (Li et al., 2026b; Qin et al., 2025) also collapse (see more details in Appendix C.1). This performance gap between VaLR-S and other latent reasoning methods provides strong evidence that latent reasoning without visual recall fails to maintain visual grounding during long reasoning traces, confirming the effectiveness of dynamic visual re-injection.

In addition, VaLR-M even achieves state-of-the-art performance (52.9%) over previous baselines, *e.g.*, GPT-4o (34.0%) and Ocean-R1 (30.5%), highlighting that the combination of different vision encoders produces the synergistic effect. In particular, VaLR-M achieves remarkable performance on spatial understanding tasks, such as relative (50.0%) and absolute (40.6%) distance prediction. These results validate our hypothesis that latent reasoning with visual alignment prevents the visual information decay observed in standard reasoning approaches.

## 4.3. Perception Tasks

We further present that VaLR also improves performance on moderate-length reasoning tasks beyond long-context by evaluating it on five perception benchmarks. We report

*Table 4.* **Ablation study for multi-encoder alignment.** Accuracy (%) for VaLR ablation of different combinations of vision encoder. We consider DINOv3 (Siméoni et al., 2025) and SigLIPv2 (Tschannen et al., 2025) for self-supervised vision encoders and $\pi^3$ (Wang et al., 2026b) for a geometry foundation model. We evaluate the model on 3D benchmark, *i.e.*, VSI-Bench (Yang et al., 2025b), and perception benchmarks including BLINK (Fu et al., 2024), MMVP (Tong et al., 2024b), MMStar (Chen et al., 2024b), V* (Wu & Xie, 2024), and CVBench (Tong et al., 2024a). Top row is a baseline, *i.e.*, Qwen2.5-VL-7B. The bold indicates the best result within the group.

| Target Encoder | | | VSI-Bench | BLINK | MMVP | MMStar | V* | CVBench |
|---|---|---|---|---|---|---|---|---|
| $\pi^3$ | DINOv3 | SigLIPv2 | | | | | | |
| ✗ | ✗ | ✗ | 33.0 | 55.7 | 56.0 | 67.1 | 76.4 | 74.5 |
| ✔ | ✔ | ✗ | 52.4 | 64.6 | 60.0 | 68.9 | 85.8 | 87.2 |
| ✔ | ✗ | ✔ | 50.5 | 63.8 | 60.0 | 70.5 | 85.3 | 87.2 |
| ✗ | ✔ | ✔ | 42.0 | 62.5 | **60.7** | 72.0 | **86.9** | 84.5 |
| ✔ | ✔ | ✔ | **52.9** | **64.7** | 60.3 | **72.3** | **86.9** | **87.6** |

accuracy on BLINK (Fu et al., 2024), MMVP (Tong et al., 2024b), MMStar (Chen et al., 2024b), V* (Wu & Xie, 2024), and CVBench (Tong et al., 2024a). We train the model with two different setups: (i) VaLR-S, which aligns a single encoder using DINOv3 (Siméoni et al., 2025) and (ii) VaLR-M, which aligns multiple encoders using DINOv3, SigLIPv2 (Tschannen et al., 2025), and $\pi^3$ (Wang et al., 2026b).

**Results.** As shown in Table 2, VaLR achieves substantial improvements over the base model. These results reveal that the learned visual grounding capability generalizes to improve short-context perception as well. The consistent advantages over other latent reasoning methods are particularly informative: VaLR-M outperforms CoVT by 8.7%p on BLINK and 8.9%p on V*, and significantly surpasses Monet and LVR across all benchmarks. Interestingly, reasoning models such as R1-OneVision and Ocean-R1 show inconsistent results, with Ocean-R1 achieving strong V* performance (78.0%) while underperforming on BLINK (56.8%) and MMVP (58.0%), suggesting that their reasoning enhancement overfit to specific task patterns rather than developing robust visual understanding. In contrast, VaLR's consistent improvements across diverse perception tasks validate that our visual alignment strategy during latent reasoning provides a general mechanism for maintaining high-quality visual representations throughout the reasoning process, regardless of reasoning length.

### 4.4. Reasoning Length Analysis

To investigate whether VaLR follows the test-time scaling law, we analyze the performance as a function of reasoning length. We consider Ocean-R1 (Lingfeng et al., 2025), and LVR (Li et al., 2026b) as baselines and evaluate on Math-Vista (Lu et al., 2024), MathVision (Wang et al., 2024a), MMhalu (Sun et al., 2024), and MMVP (Tong et al., 2024b), grouping samples by generated reasoning length to observe performance trends. Note that Ocean-R1 is a reasoning model trained with standard CoT data.

**Results.** As illustrated in Figure 2, while all baseline meth-

ods peak at intermediate reasoning lengths and subsequently degrade, VaLR shows monotonic improvement across all benchmarks. In particular, on MMVP, VaLR sustains strong performance across all reasoning lengths while Ocean-R1 dramatically collapses from 62.7% to 56.5% at 300 tokens. This divergent behavior provides compelling evidence that models trained for language reasoning or naive latent reasoning progressively lose visual priors as they generate longer reasoning chains. These results validate that VaLR successfully maintain visual grounding during extended reasoning, enabling the model to benefit from longer thinking time rather than suffer from it. We thereby achieve true test-time scaling in the multi-modal domain as widely demonstrated for language models.

*Table 5.* **Ablation study for single encoder alignment.** Accuracy (%) for VaLR ablation of varying vision foundation model including CLIP, DINOv2/v3 and SigLIPv2 during training. We consider several VQA benchmarks including BLINK, MMVP, MMStar, V*, and CVBench. The bold indicates the best result and underlined indicates the second best result within the group.

| Method | BLINK | MMVP | MMStar | V* | CVBench |
|---|---|---|---|---|---|
| Qwen2.5-VL-7B | 55.7 | 56.0 | 67.1 | 76.4 | 74.5 |
| + CLIP | 62.3 | 59.3 | 71.0 | 83.2 | 79.1 |
| + SigLIPv2 | 62.8 | 59.7 | **71.3** | 83.2 | 81.9 |
| + DINOv2 | 62.7 | 60.0 | 70.7 | 83.8 | 81.8 |
| + DINOv3 | **63.1** | **60.3** | 70.8 | **86.4** | **83.1** |

### 4.5. Ablation Study and Analysis

**Effect of representation alignment.** To verify the contribution of representation alignment (REPA) to VaLR's performance, we conduct ablation studies to test if VaLR functions effectively without external vision encoders and REPA. Specifically, during training, we replaced DINOv3 (Siméoni et al., 2025) with Qwen's native vision encoder (VaLR w/QE) and also trained VaLR without REPA (VaLR w/o VA). As shown in Table 3, VaLR trained with Qwen's native encoder still consistently outperforms other baselines even without external alignment. These results indicate that VaLR is not reliant on external vision encoders, while incorporating them further enhances the performance. Additional results are provided in Appendix C.2.

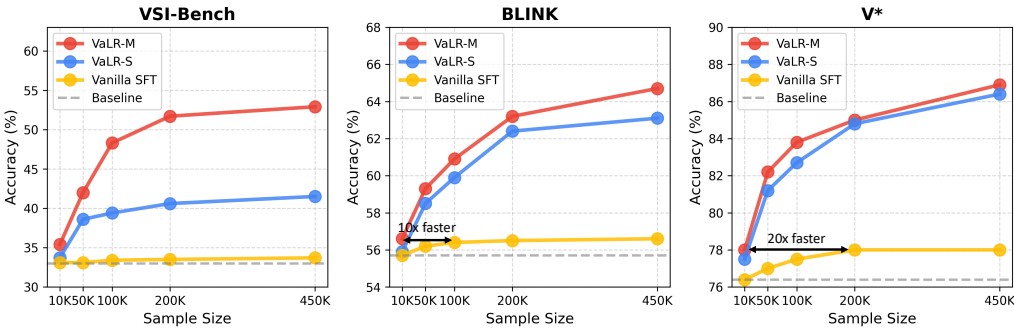

*Figure 3.* **Effect of data scalability.** We investigate the effect of the size of data and evaluate on VSI-Bench, BLINK, and V* benchmark. Results are marked 10K, 50K, 100K, 200K, and 450K sample size with fixed iterations. The result show consistent and scalable performance improvements with increased data size across all benchmarks. Notably, VaLR achieves >20x faster convergence than vanilla SFT model on V* benchmark.

**Alignment to different vision encoders.** We further analyze whether VaLR can extend to other vision encoders for representation alignment, not limited to DINOv3. Specifically, we train the Qwen2.5-VL-7B model using VaLR with the self-supervised vision encoders including CLIP (Radford et al., 2021), SigLIPv2 (Tschannen et al., 2025), and DINOv2 (Oquab et al., 2023). As shown in Table 5, VaLR consistently outperforms the base model regardless of the vision encoder choice. We observe that VaLR consistently improves performance in an encoder-agnostic manner, and yields larger gains when paired with stronger vision encoders such as DINOv3.

**Multi-encoder analysis.** To extend our analysis from the observation in Section 4.2, we investigate whether the model can align with the distinct representational characteristics of each encoder. To verify this, we conduct a control experiment on various multi-encoder variations including $\pi^3$, DINOv3, and SigLIPv2 (more details in Section 3.3). As shown in Table 4, incorporating additional encoders consistently leads to performance gains. Notably, these improvements are closely aligned with the specific characteristics of each encoder's representation. In detail, integrating the 3D-specialized encoder, $\pi^3$ significantly improves results on the 3D multi-view benchmark, VSI-Bench (Yang et al., 2025b). Moreover, adding 2D encoders, such as DINOv3 or SigLIPv2, enhances the performance across several perception benchmarks. Finally, integrating all three encoders achieves the best performance across all tasks. These results indicate that VaLR successfully aligns with distinct encoder representations by effectively leveraging their domain-specific strengths.

**Alignment layer analysis.** We investigate which intermediate layer of MLLMs is most effective for alignment via REPA. Specifically, we vary the layer index across three settings—Front (4th), Middle (12th), and Last (27th). As shown in Table 6, while all settings improve performance, REPA applied at the middle layer achieves the strongest

*Table 6.* **Ablation study for alignment layer.** Accuracy (%) for VaLR ablation of different alignment layer of MLLM. We consider perception benchmarks consist with BLINK, MMVP, MMStar, V*, and CVBench. Front, Middle, and Last denotes 4, 12, 27-*th* layer index in Qwen2.5-VL-7B, respectively.

| Method | BLINK | MMVP | MMStar | V* | CVBench |
|---|---|---|---|---|---|
| Qwen2.5-VL-7B | 55.7 | 56.0 | 67.1 | 76.4 | 74.5 |
| Front | 59.2 | 55.7 | 68.5 | 83.8 | 78.6 |
| Middle | 63.1 | 60.3 | 70.8 | 86.4 | 83.1 |
| Last | 62.8 | 60.0 | 70.8 | 85.3 | 82.5 |

results. This observation is consistent with prior studies (Yu et al., 2025b; Kang et al., 2025; Jiang et al., 2025) indicating that visual information is most prominently represented in the middle layers of MLLMs.

**Data Scalability.** We investigate the data scalability of VaLR by tracking the performance of the checkpoints trained on different numbers of samples, *e.g.*, 10K, 50K, 100K, 200K, and 450K. We compare the performance of Vanilla-SFT, VaLR-S, and VaLR-M on three benchmarks: VSI-Bench, BLINK, and V*. As shown in Figure 3, both VaLR variants consistently improve performance as the sample size grows, while Vanilla-SFT saturates beyond 200K samples. Notably, our best model VaLR-M achieves $> 20\times$ faster training to reach comparable performance on V*. This suggests that aligning with encoders facilitates learning richer features from the training data and improves data scalability.

# 5. Conclusion

In this paper, we have presented VaLR, a multi-modal reasoning framework that generates vision-aligned latent tokens during the reasoning process. Our experiments showed that VaLR performs test-time scaling behavior and consistently improves performance on various benchmarks that require a long or short context. We hope our work will facilitate future research on reasoning in multi-modal large language models.

## Impact Statement

Recent advancements of Multi-modal Large Language Models (MLLMs) have enabled remarkable performance in vision question answering. However, these models suffer from dilution of visual information during autoregressive text generation. This phenomenon is emphasized during long-context reasoning. Consequently, it prevents the use of MLLM in domains that require long-context reasoning, such as the Vision Language Action (VLA) model and the Computer Use Agent (CUA).

Our work addresses these challenges by proposing an effective way to inject a visual checkpoint with a latent token. Our approach improves long-context reasoning with test-time scalability and general VQA performance. We believe VaLR suggests future directions for mitigating the dilution of vision information.

In context of applications, alleviating the long-context reasoning reveals the use of MLLM in much more complex tasks, especially when visual information is involved, such as a robot with VLA or proactive CUA. This automatic agentic system can facilitate the innovation of human society.

## Acknowledgments

This work was partly supported by Institute for Information & Communications Technology Planning & Evaluation (IITP) grant funded by the Korea government (MSIT) (RS-2019-II190075, Artificial Intelligence Graduate School Program (KAIST); RS-2022-II220959, Few-shot Learning of Causal Inference in Vision and Language for Decision Making; RS-2024-00509279, Global AI Frontier Lab; RS-2025-25442469, Development of Edge AI Server Technology with High-Performance and High-Reliability, 33%), and RLWRLD Inc. We also thank Min-Hung Chen and Ryo Hachiuma for their valuable comments and suggestions in preparing an earlier version of the manuscript.

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

# A. Implementation Details

## A.1. Training Details

We adopt Qwen2.5-VL-7B (Bai et al., 2025) as the base model and perform the supervised fine-tuning. In stage 1, we freeze the vision encoder and train only the language model backbone. In stage 2, we continue to freeze the vision encoder while jointly training the language model and the MLP for alignment. Detailed hyperparameters are provided in Table 7. In both stages, we train the model with the same Chain-of-Thought (CoT) datasets. Details of dataset construction are provided in the Appendix B. All experiments are conducted with 4x NVIDIA Tesla A100s.

*Table 7.* **Hyperparameters for Stage 1 and 2.**

| Hyperparameter | Stage 1 | Stage 2 |
|---|---|---|
| optimizer | AdamW | |
| deepspeed | Zero-2 | |
| learning rate | 1e-5 | 2e-6 |
| MLP $\psi$ learning rate | - | 1e-5 |
| per-GPU batch size | 2 | |
| gradient accumulation steps | 16 | |
| weight decay | 0.01 | |
| epoch | 1 | |
| warm-up ratio | 0.03 | |
| latent tokens ($K$) | - | 16 |
| alignment weight ($\lambda$) | - | 0.5 |

During training, we select CLIP (Radford et al., 2021), SigLIP (Tschannen et al., 2025), DINO (Oquab et al., 2023; Siméoni et al., 2025) and $\pi^3$ (Wang et al., 2026b) for target of representation alignment. All vision encoders are ViT-L.

## A.2. Evaluation Details

We use VLMEvalKit (Duan et al., 2024) for all evaluations. We adapt LLM-as-a-Judge in our evaluation process and use GPT-4o (Hurst et al., 2024) as the judge. To evaluate Monet (Wang et al., 2026a), we follow the system prompt proposed by the Monet authors. For CoVT (Qin et al., 2025), we use CoVT-7B-depth-seg-dino. We evaluate various models on VSI-Bench (Yang et al., 2025b) for 3D spatial reasoning tasks, BLINK (Fu et al., 2024), MMVP (Tong et al., 2024b), MMStar (Chen et al., 2024b), V* (Wu & Xie, 2024), and CVBench (Tong et al., 2024a) for perception tasks, and MathVista (Lu et al., 2024), MathVision (Wang et al., 2024a), and MMhalu (Sun et al., 2024) for reasoning length-wise analysis. We follow the evaluation protocol specified by each benchmark. Specifically, for VSI-Bench, we select the number of frames for multi-view images as summarized in Table 8. In addition, we report the model versions used for API-based evaluation as follows:

- `openai/gpt-4o-2024-08-06`
- `Claude/claude-sonnet-4-20250514`

*Table 8.* **Number of frames used in VSI-Bench evaluation.**

| Methods | # of Frames |
|---|---|
| GPT-4o | 16 |
| LLaVA-NeXT-Video-7B | 32 |
| R1-OneVision-7B | 32 |
| Ocean-R1-7B | 32 |
| Qwen2.5-VL-7B | 32 |
| LVR | 32 |
| CoVT | 32 |
| Monet | 32 |
| VaLR (Ours) | 32 |

# B. Dataset Construction

To train VaLR, we use a collection of existing Chain-of-Thought (CoT) datasets including interleaved and non-interleaved datasets. We provide detailed statistics for our datasets in Appendix B.1. To enable latent reasoning, we tailor the existing datasets following the procedure outlined in the main paper. We elaborate more details for non-interleaved dataset in Appendix B.2, and for interleaved dataset in Appendix B.3.

## B.1. Data Statistics

We collect 450K samples from the mixture of several open-source datasets. Specifically, for the 125K interleaved CoT samples, we select Zebra-CoT (Li et al., 2026a), CogCoM (Qi et al., 2025), ReFocus (Fu et al., 2025), and Visual-CoT (Shao et al., 2024a). For the remaining 325K non-interleaved CoT data, we choose GCoT (Chen et al., 2026) with 170K samples from OneThinker-SFT (Feng et al., 2026).

## B.2. Non-interleaved CoT Data

Let an input image set be $\mathcal{I} = \{I_1, \cdots, I_Q\}$ where $Q$ is the number of input images, and the ground-truth language CoT reasoning be $\mathbf{y} = [r^1, r^2, \cdots, r^N, a]$ where $r^i$ is the $i$-th reasoning step and $a$ is the final answer.

**Single-view VQA dataset.** For single-view data where only one input image is given, we extract a visual representation from the input image $I_1$ using a vision encoder. Before the reasoning step $r^1$, the model learns latent token generation with representation alignment (REPA) between the MLLM and the extracted features. This ensures that the latent reasoning tokens are grounded in the visual features from the beginning of the reasoning process.

**Multi-view VQA dataset.** For CoT data with multiple input images, standard CoT datasets often do not explicitly specify which image should be recalled at each reasoning step. To address this issue, we employ GPT-4o (Hurst et al., 2024) to identify which image is most relevant for each reasoning step $r^{(i)}$ in the ground-truth CoT reasoning. Specifically, we process GPT-4o with the set of input images $\mathcal{I}$ and the CoT reasoning chain $\mathbf{y}$, and ask it to match each reasoning step with its corresponding target image. After obtaining the target image $I_{\text{target}}$ for each reasoning step $r^{(i)}$, we apply REPA in the same manner as for single-view data, aligning the latent tokens with the visual features of the identified target image. The prompt used for multi-view data curation is provided below.

---

**Prompt for Multi-view Data Curation**

This is a Chain-of-Thought (CoT) VQA data including multiple input images and corresponding CoT. Please divide the CoT step-by-step. Then find a useful and proper target image for each step. Lastly, place the target image in front of each reasoning step.

---

## B.3. Interleaved CoT Data

Let $I_i$ be an $i$-th initial input image and $I_i^{\text{inter}}$ be an interleaved input image. The input image set can be expressed as $\mathcal{I} = \{I_1, \cdots, I_Q, I_1^{\text{inter}}, \cdots, I_R^{\text{inter}}\}$ where $Q$ and $R$ be the number of initial input image and interleaved image, respectively. Additionally, let the ground-truth language CoT reasoning be $\mathbf{y} = [r^1, r^2, \cdots, r^N, a]$ where $r^j$ is the $j$-th reasoning step and $a$ is the final answer.

For interleaved data, images are inserted at specific positions within the CoT reasoning process. These interleaved images naturally indicate critical points in reasoning where visual information is needed. Therefore, we initiate the latent mode at the position where each interleaved image appears. Specifically, when an interleaved image $I_i^{\text{inter}}$ is encountered before each reasoning step $r^j$, we use the image as the target image for representation alignment. This approach allows the model to dynamically recall and integrate visual information at the exact moments specified by the dataset, effectively leveraging the explicit visual checkpoints provided in interleaved CoT data. By aligning the latent tokens with the features from $I_i^{\text{inter}}$ at these designated positions, VaLR learns to naturally incorporate visual information when transitioning between reasoning steps.

# C. Additional Experimental Results

## C.1. Detailed Analysis in Table 1

In Table 1, we observe that latent reasoning baselines including LVR (Li et al., 2026b), CoVT (Qin et al., 2025), and Monet (Wang et al., 2026a) collapse on the multi-view benchmark (Yang et al., 2025b). This occurs because existing approaches can only cover single-view scenarios or have limited extension to multi-view settings. As a result, these baselines show vulnerable performance on tasks requiring long-term visual memory. In contrast, our method, VaLR, is applicable to both single-view and multi-view scenarios and demonstrates a robust framework for long-context reasoning, achieving state-of-the-art performance on the multi-view benchmark.

## C.2. Detailed Analysis in Table 4

In Table 4, we report representation alignment (REPA) results for external vision encoders, *e.g.*, $\pi^3$ (Wang et al., 2026b), DINOv3 (Siméoni et al., 2025), and SigLIPv2 (Tschannen et al., 2025). In Table 9, we additionally apply REPA to the base model's native encoder, *i.e.*, Qwen's encoder. Similar to the findings in Table 4, the model that aligns with all encoders including Qwen's encoder achieves the best overall performance, demonstrating the potential benefit obtained from additional alignment with stronger encoders.

*Table 9.* **Detailed ablation study for multi-encoder alignment.** Accuracy (%) for VaLR ablation of different combinations of vision encoder. We consider $\pi^3$ for a geometry foundation model, DINOv3 and SigLIPv2 for self-supervised vision encoder, and Qwen encoder. We evaluate the model on multi-view benchmark, *i.e.*, VSI-Bench, and perception benchmarks including BLINK, MMVP, MMStar, V$^*$, and CVBench. Top row is a baseline, *i.e.*, Qwen2.5-VL-7B. The bold indicates the best result within the group.

| Target Encoder | | | | VSI-Bench | BLINK | MMVP | MMStar | V$^*$ | CVBench |
|---|---|---|---|---|---|---|---|---|---|
| $\pi^3$ | DINOv3 | SigLIPv2 | Qwen Enc. | | | | | | |
| ✗ | ✗ | ✗ | ✗ | 33.0 | 55.7 | 56.0 | 67.1 | 76.4 | 74.5 |
| ✔ | ✔ | ✗ | ✗ | 52.4 | 64.6 | 60.0 | 68.9 | 85.8 | 87.2 |
| ✔ | ✗ | ✔ | ✗ | 50.5 | 63.8 | 60.0 | 70.5 | 85.3 | 87.2 |
| ✔ | ✗ | ✗ | ✔ | 51.6 | 63.5 | 60.3 | 67.6 | 84.7 | 85.9 |
| ✗ | ✔ | ✔ | ✗ | 42.0 | 62.5 | 60.7 | 72.0 | 86.9 | 84.5 |
| ✗ | ✔ | ✗ | ✔ | 41.4 | 63.4 | 60.3 | 71.1 | 87.4 | 85.0 |
| ✗ | ✗ | ✔ | ✔ | 40.9 | 60.7 | **61.3** | 72.3 | 84.4 | 83.0 |
| ✔ | ✔ | ✔ | ✗ | 52.9 | 64.7 | 60.3 | 72.3 | 86.9 | **87.6** |
| ✗ | ✔ | ✔ | ✔ | 41.7 | 63.1 | 61.0 | **72.6** | 88.0 | 86.0 |
| ✔ | ✔ | ✔ | ✔ | **53.3** | **65.1** | 61.0 | 72.3 | **88.5** | 87.4 |

## C.3. Clock-time Reasoning Analysis

In this section, we analyze the computational cost incurred by the proposed additional components. We measure clock-time with batch size 1 on a single NVIDIA Tesla A100 GPU using vLLM (Kwon et al., 2023). We evaluate LVR (Li et al., 2026b), Monet (Wang et al., 2026a), and VaLR on 32-view and 1-view scenarios from VSI-Bench (Yang et al., 2025b) and CVBench (Tong et al., 2024a), respectively.

*Table 10.* **Clock-time reasoning analysis.**

| Method | 32-view | 1-view |
|---|---|---|
| Qwen2.5-VL | 1.21 | 0.64 |
| + vanilla SFT | 1.43 | 0.68 |
| + LVR | 1.49 | 0.66 |
| + Monet | 1.51 | 0.79 |
| +VaLR (Ours) | 1.55 | 0.80 |

## C.4. REPA vs. Input Tokens for Visual Representations

We investigate how effectively our method utilizes external vision encoders. Specifically, we compare our representation alignment (REPA) approach with the method using DINOv3 (Siméoni et al., 2025) features as input tokens to the LLM backbone. As shown in Figure 4, REPA outperforms the input token method on VSI-Bench (Yang et al., 2025b), BLINK (Fu et al., 2024), and V* (Wu & Xie, 2024) benchmark. In particular, unlike the input token method, VaLR does not require vision encoder at test-time, indicating that our method is highly efficient and effective.

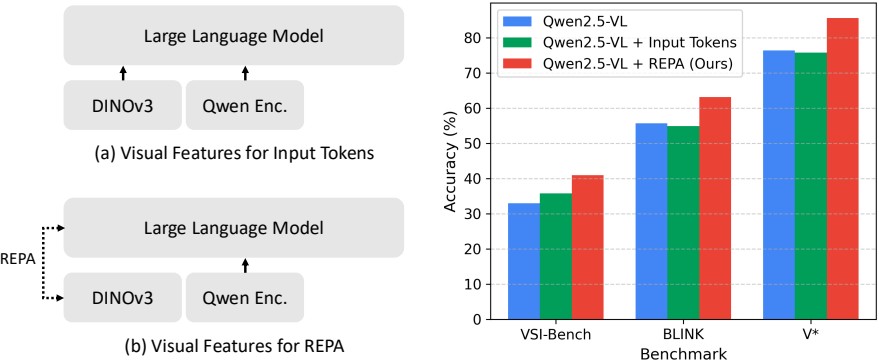

*Figure 4.* **Comparison between methods using visual representations.** We compare two methods using DINOv3 features: (a) Using visual features as input visual tokens of MLLM (Green), (b) Aligning visual features with MLLM embeddings (Red). We report accuracy (%) on VSI-Bench, BLINK, and V* benchmark.

## C.5. Feature Visualization

We visualize the changes in MLLM intermediate features through representation alignment. Features of VaLR are extracted from 12-th layer.

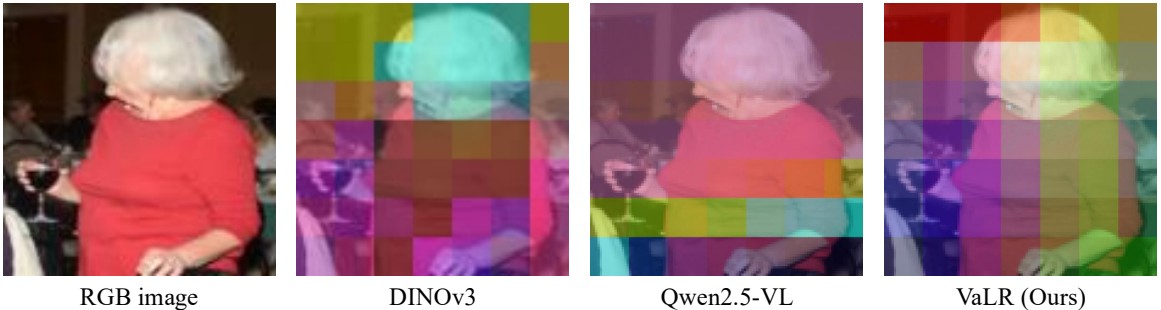

RGB image      DINOv3      Qwen2.5-VL      VaLR (Ours)

*Figure 5.* **Feature visualization.**

## C.6. Additional Ablation Study

We provide additional ablation studies for VaLR. All experiments are conducted on the aligned model with a single encoder (Siméoni et al., 2025).

**Ablation study for $\lambda$.** During training, we use the standard autoregressive loss and the representation alignment (REPA) loss, where $\lambda$ is the weight for the REPA loss. We conduct an ablation study to examine the effect of $\lambda$. As shown in Table 11, VaLR achieves the best performance at $\lambda = 0.5$, demonstrating a good balance where language semantics are preserved while maintaining visual alignment.

*Table 11.* **Ablation study for $\lambda$.**

| $\lambda$ | VSI-Bench | BLINK | V$^*$ |
|------|-----------|-------|------|
| 0.0  | 34.0 | 57.1 | 75.9 |
| 0.25 | 38.5 | 61.7 | 78.5 |
| 0.5  | 41.5 | 63.1 | 86.4 |
| 0.75 | 40.4 | 62.8 | 84.8 |
| 1.0  | 40.0 | 60.9 | 84.3 |

**Ablation study for $K$.** We investigate the effect of the number of latent tokens $K$, which determines the resolution of vision-aligned features in the MLLM's latent mode. As shown in Table 12, we observe that increasing the number of latent tokens has clear advantage.

*Table 12.* **Ablation study for $K$.**

| $K$ | VSI-Bench | BLINK | V$^*$ |
|-----|-----------|-------|------|
| 1   | 33.9 | 58.6 | 78.0 |
| 4   | 34.5 | 62.0 | 85.3 |
| 9   | 39.7 | 62.8 | 85.3 |
| 16  | 41.5 | 63.1 | 86.4 |
| 25  | 41.7 | 63.2 | 87.4 |

