# OpenReview forum: "Vision-aligned Latent Reasoning for Multi-modal Large Language Model"
_ICML.cc/2026/Conference — ICML 2026 regular_

### Official Review · Reviewer_sya1 · 2026-03-03

**Soundness:** 3
**Presentation:** 3
**Significance:** 3
**Originality:** 2
**Overall Recommendation:** 5
**Confidence:** 4

**Summary:**

The paper might be inspired by the REPA for the diffusion model training, proposing the same idea for the vlms, in the thinking task.
The paper aims to align the visual latents from the intermediate layers of the language models to the representations from the 3rd-party vision encoders, such as dino or siglip.
The goal is to solve the degradation problems of the vision signal in the case of long context thinking prediction.

**Compliance With Llm Reviewing Policy:**

Affirmed.

**Final Justification:**

The rebuttals address my concerns.

**Key Questions For Authors:**

See above.

**Limitations:**

The paper didn't discuss the limitation, the main limitation could be about the memory especially under the situation fo training large vlms with long-context window.

**Strengths And Weaknesses:**

The method is simple and effective, which I like it. Also the experiments are solid, with a quite large improvements on each benchmark for the long-context reasoning task.
Aslo the paper has solid ablations and visualizations that I want to see. Specifically, the experiments of Table 1 shows that the baseline models collapsed on the interleaved reasoning task, but the proposed the model can handle it with long-context reasoning.
Almostly, I don't have many questions but only two:

for caching so many 3rd party vision encoders for the repa loss, how much the memory cost for training in the experiments, especially for the interleaved reasoning traces. It would be good to show the cost for readers. If the memory is limited, it is hard to use this method to train vlms, especially with long context window.

for the model size, the paper only conducts the main experiments on 7B model, we all know, the larger language or vision-language model can dramatically have robust representation for the tasks. So the concern is that if scaling up the base model size like from 7B to 13B or even larger, the method effectiveness with scaling up model size is not clear in the paper. If the method still works well with scaling up the model size, it could make the method more solid. Qwen-vl might not have the middle model size between 7B and 32B, but other sota vl model family might have like 14B or else.

---

is this a type in the abstract? the p in ``19.9%p gain''

---

> ### Author Rebuttal · Authors · 2026-03-31
>
> Dear Reviewer sya1,
>
> We sincerely appreciate your valuable comments and efforts. We do our best to clarify each of the concerns carefully as follows:
>
> ---
>
> **[Q1, L1] Memory overhead from multiple vision encoders during training**
>
> We agree that memory overhead is an important practical concern. In our implementation, we adopt an offline feature precomputation strategy: the visual features from all vision encoders are precomputed and cached prior to training, so that no additional encoder needs to be loaded into GPU memory during the actual training process. As a result, the additional GPU memory introduced by our method amounts only to a shallow MLP per encoder for alignment, which scales negligibly with the number of encoders and imposes no meaningful constraint even in memory-limited environments. We will make this implementation detail more explicit in the revised manuscript.
>
> ---
>
> **[Q2] Generality of method in different model scales**
>
> To validate the generalizability of our method across different model scales and MLLM backbones, we conducted additional experiments applying VaLR to a broader set of models, including Qwen3-VL and InternVL-3.5 at multiple scales (2B, 8B, and 14B).
>
> As shown in Table below, the 2B model with VaLR achieves performance comparable to the pre-trained 14B model, demonstrating that VaLR is effective even with robust representations from larger models, indicating that our method scales well with model size.
>
> Furthermore, VaLR consistently improves performance on different backbones. For instance, Qwen3-VL-8B gains +7.1\%p and InternVL-3.5-14B gains +3.2\%p in VSI-Bench. This suggests that VaLR is well generalized to various MLLMs.
>
> |  | VSI-Bench | BLINK | MMVP | MMStar | V\* | CVBench |
> | :---- | :---- | :---- | :---- | :---- | :---- | :---- |
> | Qwen2.5-VL-7B | 33.0 | 55.7 | 56.0 | 67.1 | 76.4 | 74.5 |
> | \+Ours | **52.9** | **64.7** | **60.3** | **72.3** | **86.9** | **87.6** |
> | Qwen3-VL-8B | 59.4 | 69.1 | 59.0 | 70.9 | 90.1 | 82.3 |
> | \+Ours | **66.5** | **70.8** | **67.3** | **77.4** | **92.1** | **91.7** |
> | InternVL-3.5-2B | 53.8 | 51.3 | 54.0 | 62.7 | 80.6 | 73.6 |
> | \+Ours | **60.5** | **62.4** | **61.0** | **70.2** | **85.8** | **87.7** |
> | InternVL-3.5-8B | 56.3 | 59.5 | 61.7 | 69.3 | 83.2 | 77.1 |
> | \+Ours | **62.1** | **66.9** | **68.7** | **76.8** | **90.1** | **88.2** |
> | InternVL-3.5-14B | 60.8 | 57.6 | 63.7 | 70.4 | 82.7 | 80.0 |
> | \+Ours | **64.0** | **69.7** | **70.3** | **77.7** | **91.6** | **90.9** |
>
> ---
>
> **[Q3] Typo**
>
> We appreciate the careful reading. "%p" denotes "percentage points," a standard notation used to express absolute differences between two percentages, avoiding ambiguity with relative percentage changes. For instance, an improvement from 33.0% to 52.9% corresponds to a 19.9%p gain.

---

> > ### Author Rebuttal · Reviewer_sya1 · 2026-03-31
> >
> > My main concern about the method with scaling VLMs is solved in rebuttal, and I also have checked the other reviewers' comments and responses for them. I would love to maintain the rating as A.

---

> > > ### Author Response · Authors · 2026-04-04
> > >
> > > Dear Reviewer sya1,
> > >
> > > Thank you for your comments and for taking the time to review our manuscript. We are pleased to hear that our replies have resolved your concerns.
> > >
> > > If you have any further comments or remaining issues, please let us know. In particular, if there is a main concern that we should address for you to consider raising our score, we value your feedback and will provide a response to it.
> > >
> > > Best regards,\
> > > Authors

---

### Official Review · Reviewer_2z6E · 2026-03-12

**Soundness:** 3
**Presentation:** 3
**Significance:** 3
**Originality:** 3
**Overall Recommendation:** 4
**Confidence:** 4

**Summary:**

This paper proposes VaLR, a novel framework to enhance the multi-step reasoning capabilities of MLLLMs. The core method is to combine repa into the latent reasoning process and introduce a fixed number of vision-aligned latent tokens before each reasoning step. Empirical results demonstrate the effectiveness of VaLR on general tasks over other latent reasoning methods.

**Compliance With Llm Reviewing Policy:**

Affirmed.

**Final Justification:**

The rebuttal has resolved my concerns.

**Key Questions For Authors:**

1. How much of the inference cost is introduced by VaLR? Could the authors provide a quantitative analysis of the trade-off between K and inference latency?
2. Can VaLR work on more recent baselines, such as Qwen3-VL?

**Limitations:**

Scaling behaviour of VaLR when applied to large VLM models with much more pretrained information.

**Strengths And Weaknesses:**

Strength:
1. The experimental parts are thorough and extensive. The results are generally convincing. The author provides results on many benchmarks and includes sufficient ablations.
2. The proposed method is well-motivated and simple. The core innovation is to incorporate repa with selected external vision encoders, such as DINOv3 or SigLIPv2, to strengthen the latent understanding of visual information in the latent LLM space. While this method is straightforward, its effectiveness and simplicity are appreciated.
3. The writing of this paper is clear and easy to follow.

Weakness:
1. The author claims that this method is superior to other latent visual models for addressing the problem of  "gradual loss of visual context". Can this be supported by specific experiments rather than common benchmarks?
2. The use of cosine similarity for vision alignment is quite trivial. There are no ablations about this loss function. What about MSE loss, or other similarity losses for this part? Is cosine similarity the only effective one?
3. In Table 6, ablations indicate that the middle layers' features are best for alignment. This finding may be model-specific. When applied to larger or smaller models, does this finding hold true for them?

---

> ### Author Rebuttal · Authors · 2026-03-31
>
> Dear Reviewer 2z6E,
>
> We sincerely appreciate your valuable comments and efforts. We do our best to clarify each of the concerns carefully as follows:
>
> ---
> **[W1] Experimental evidence for visual dilution in MLLMs**
>
> We analyze visual dilution using the Prompt Dependency Measure (PDM) [1] following prior work [1,2,3], which quantifies dependence on visual input. As shown in the table below, both base models exhibit a clear decline in PDM as generation progresses, indicating progressive visual dilution. In contrast, VaLR maintains a substantially higher and more stable PDM throughout generation. For example, Qwen2.5-VL's PDM drops by 63\% from length 50 to 200 (0.38 $\rightarrow$ 0.14), and Qwen3-VL's drops by 49\% (0.41 $\rightarrow$ 0.21). With VaLR, these drops are reduced to only 15\% (0.41 $\rightarrow$ 0.35) and 9\% (0.46 $\rightarrow$ 0.42), respectively. This directly supports our claim that VaLR effectively mitigates visual dilution.
>
> |  | 50 | 100 | 150 | 200 |
> | :---- | :---- | :---- | :---- | :---- |
> | Qwen2.5-VL | 0.38 | 0.28 | 0.20 | 0.14 |
> | +Ours | **0.41** | **0.39** | **0.37** | **0.35** |
> | Qwen3-VL | 0.41 | 0.34 | 0.28 | 0.21 |
> | +Ours | **0.46** | **0.44** | **0.43** | **0.42** |
>
> Furthermore, we will include additional qualitative analysis based on attention score distributions in the revised manuscript.
>
> \
> [1] Favero, Alessandro, et al. "Multi-modal hallucination control by visual information grounding." CVPR 2024.
>
> [2] Zou, Xin, et al. "Look twice before you answer: Memory-space visual retracing for hallucination mitigation in multimodal large language models." arXiv (2024).
>
> [3] Jung, Mingi, et al. "Visual attention never fades: Selective progressive attention recalibration for detailed image captioning in multimodal large language models." arXiv (2025).
>
> ---
>
> **[W2] Lack of ablation study for alignment loss**
>
> We additionally conducted an ablation study on the alignment loss function. Specifically, we compare cosine similarity against Mean Squared Error (MSE) and Normalized Temperature-scaled Cross Entropy (NT-Xent) in both single-encoder and multi-encoder alignment settings. As shown in the table below, cosine similarity achieves the best performance in both settings. Notably, MSE performs less effectively in the multi-encoder setting, likely because it is sensitive to differences in feature scale and norm across encoders, whereas cosine similarity naturally normalizes these variations by operating on directional similarity alone.
>
> |  | VSI-Bench | BLINK | V\* |
> | ----- | :---- | :---- | :---- |
> | *Single-encoder alignment* |  |  |  |
> | MSE | 39.9 | 62.6 | 85.8 |
> | NT-Xent | 41.1 | 61.9 | 84.8 |
> | Cos Sim. (ours) | 41.5 | 63.1 | 86.4 |
> | *Multi-encoder alignment* |  |  |  |
> | MSE | 47.4 | 63.0 | 84.3 |
> | NT-Xent | 52.2 | 63.1 | 86.3 |
> | Cos Sim. (ours) | 52.9 | 64.7 | 86.9 |
>
> ---
>
> **[W3] Analysis on alignment target layer in model scaling**
>
> To verify the generalizability of layer selection, we trained the model with different scales. As shown in the table below, middle-layer alignment consistently achieves the best performance across all model scales, suggesting that the middle layer alignment is not model-specific.
>
> | Method | BLINK | MMVP | MMStar | V\* | CVBench |
> | :---- | :---- | :---- | :---- | :---- | :---- |
> | InternVL-3.5-2B | 51.3 | 54.0 | 62.7 | 80.6 | 73.6 |
> | Front (4th) | 57.2 | 53.3 | 65.4 | 79.1 | 74.8 |
> | Middle (12th) | **62.4** | **61.0** | **70.2** | **85.8** | **87.7** |
> | Last (27th) | 61.9 | 59.7 | 70.0 | 84.3 | 85.9 |
> | InternVL-3.5-8B | 59.5 | 61.7 | 69.3 | 83.2 | 77.1 |
> | Front (6th) | 61.8 | 62.7 | 71.7 | 83.2 | 81.4 |
> | Middle (16th) | **66.9** | **68.7** | **76.8** | **90.1** | **88.2** |
> | Last (27th) | 64.2 | 68.3 | 76.1 | 88.5 | 86.6 |
> | InternVL-3.5-14B | 57.6 | 63.7 | 70.4 | 82.7 | 80.0 |
> | Front (8th) | 60.0 | 65.3 | 71.5 | 83.8 | 81.9 |
> | Middle (18th) | **69.7** | **70.3** | **77.7** | **91.6** | **90.9** |
> | Last (39th) | 66.5 | 69.3 | 76.7 | 89.5 | 88.7 |
>
> *Front (K) denotes K-th layer*
>
> ---
>
> **[Q1] Analysis of trade-off between K and inference latency**
>
> As reported in Table 10 of our manuscript, we provided the inference cost of VaLR. We further analyze how inference latency scales with K. As shown in the table below, K=16 introduces an additional 340ms inference cost. Furthermore, this increased inference cost is accompanied by consistent performance gains on VSI-Bench, rising from 33.0\% to 41.5\% as K increases from 0 to 16, demonstrating a favorable trade-off between latency and accuracy.
>
> | Method | \# of latent tokens (K) | 32-view (sec) | VSI-Bench (\%) |
> | :---- | :---- | :---- | :---- |
> | Qwen2.5-VL | \- | 1.21 | 33.0 |
> | \+Ours | 4 | 1.40 | 34.5 |
> |  | 9 | 1.44 | 39.7 |
> |  | 16 | 1.55 | 41.5 |
> |  | 25 | 1.69 | 41.7 |
>
> ---
>
> **[Q2, L1] Generality of method in recent MLLM backbones and model scales**
>
> VaRL can be generalized to various MLLM backbones and model scales. We refer to our response for Reviewer **sya1 [Q2]**.

---

> > ### Author Rebuttal · Reviewer_2z6E · 2026-04-03
> >
> > Thanks for your response. I keep my positive score.

---

> > > ### Author Response · Authors · 2026-04-04
> > >
> > > Dear Reviewer 2z6E,
> > >
> > > Thank you for your comments and for taking the time to review our manuscript. We are pleased to hear that our replies have resolved your concerns.
> > >
> > > If you have any further comments or remaining issues, please let us know. In particular, if there is a main concern that we should address for you to consider raising our score, we value your feedback and will provide a response to it.
> > >
> > > Best regards,\
> > > Authors

---

### Official Review · Reviewer_4kex · 2026-03-13

**Soundness:** 3
**Presentation:** 3
**Significance:** 3
**Originality:** 3
**Overall Recommendation:** 4
**Confidence:** 3

**Summary:**

This paper proposes VaLR, a method to enhance multi-modal reasoning by preventing visual information loss during long reasoning chains. It inserts latent tokens before each reasoning step and aligns them with features from external vision encoders. Experiments on Qwen2.5-VL show significant improvements on VQA benchmarks, particularly on long-context tasks like VSI-Bench. The approach demonstrates effective test-time scaling with increasing reasoning length.

**Compliance With Llm Reviewing Policy:**

Affirmed.

**Key Questions For Authors:**

see weakness

**Limitations:**

yes

**Strengths And Weaknesses:**

Strengths

· Addresses an important and underexplored problem of visual information decay in long reasoning chains

· Proposes a technically elegant and empirically effective latent token alignment method

· Demonstrates comprehensive and consistent improvements across multiple benchmarks

· Provides compelling evidence of maintained visual grounding through test-time scaling analysis

Weaknesses

· Training computational costs are not thoroughly analyzed

· Limited evaluation on a single base model architecture restricts claims of general applicability

· GPT-4o-based data curation for multi-view alignment may introduce additional bias and overhead

---

> ### Author Rebuttal · Authors · 2026-03-31
>
> Dear Reviewer 4kex,
>
> We sincerely appreciate your valuable comments and efforts. We do our best to clarify each of the concerns carefully as follows:
>
> ---
>
> **[W1] Lack of computational cost analysis during training time**
>
> We provide an analysis of the computational costs during training. As shown in the table below, VaLR-S (single-encoder alignment) and VaLR-M (multi-encoder alignment) require only 3.6 and 5.1 additional hours of training compared to the baseline, respectively, while achieving substantial performance improvements on VSI-Bench (33.0 $\rightarrow$ 41.5 for VaLR-S and 33.0 $\rightarrow$ 52.9 for VaLR-M). We believe this represents a highly favorable trade-off between training cost and performance gain.
>
> |  | Training time (hr) | VSI-Bench |
> | :---- | :---- | :---- |
> | Qwen2.5-VL | 28.6 | 33.0 |
> | \+VaLR-S | 32.2 | 41.5 |
> | \+VaLR-M | 33.7 | 52.9 |
>
> ---
>
> **[W2] General applicability of proposed method**
>
> To address generalizability of our method across different MLLM backbones and model scales, we conducted additional experiments applying VaLR to a broader set of models, including Qwen3-VL and InternVL-3.5 at multiple scales (2B, 8B, and 14B).
>
> As shown in Table below, VaLR consistently improves performance on different backbones and model scales. For instance, Qwen3-VL-8B gains +7.1\%p and InternVL-3.5-14B gains +3.2\%p in VSI-Bench. This suggests that VaLR is well generalized to various MLLMs.
>
> Notably, the 2B model with VaLR achieves performance comparable to the pre-trained 14B model, demonstrating that VaLR is effective even with robust representations from larger models, indicating that our method scales well with model size.
>
> |  | VSI-Bench | BLINK | MMVP | MMStar | V\* | CVBench |
> | :---- | :---- | :---- | :---- | :---- | :---- | :---- |
> | Qwen2.5-VL-7B | 33.0 | 55.7 | 56.0 | 67.1 | 76.4 | 74.5 |
> | \+Ours | **52.9** | **64.7** | **60.3** | **72.3** | **86.9** | **87.6** |
> | Qwen3-VL-8B | 59.4 | 69.1 | 59.0 | 70.9 | 90.1 | 82.3 |
> | \+Ours | **66.5** | **70.8** | **67.3** | **77.4** | **92.1** | **91.7** |
> | InternVL-3.5-2B | 53.8 | 51.3 | 54.0 | 62.7 | 80.6 | 73.6 |
> | \+Ours | **60.5** | **62.4** | **61.0** | **70.2** | **85.8** | **87.7** |
> | InternVL-3.5-8B | 56.3 | 59.5 | 61.7 | 69.3 | 83.2 | 77.1 |
> | \+Ours | **62.1** | **66.9** | **68.7** | **76.8** | **90.1** | **88.2** |
> | InternVL-3.5-14B | 60.8 | 57.6 | 63.7 | 70.4 | 82.7 | 80.0 |
> | \+Ours | **64.0** | **69.7** | **70.3** | **77.7** | **91.6** | **90.9** |
>
> ---
>
> **[W3] Bias from GPT-based curation in multi-view reasoning data**
>
> To verify that GPT-based curation is robust to potential bias, we train VaLR-M on fixed 100K multi-view samples using ground-truth annotations and GPT-curated annotations respectively, and compare their performance. As shown in the table below, the performance gap between the two is negligible, demonstrating that the bias introduced by GPT is practically marginal in VaLR training.
>
> |  | VSI-Bench | BLINK |
> | :---- | :---- | :---- |
> | Ground-truth | 48.7 | 60.8 |
> | GPT-curated | 48.6 | 60.9 |

---

> > ### Author Rebuttal · Reviewer_4kex · 2026-04-03
> >
> > Thank you for your effort, I'll keep my score.

---

> > > ### Author Response · Authors · 2026-04-04
> > >
> > > Dear Reviewer 4kex,
> > >
> > > Thank you for your comments and for taking the time to review our manuscript. We are pleased to hear that our replies have resolved your concerns.
> > >
> > > If you have any further comments or remaining issues, please let us know. In particular, if there is a main concern that we should address for you to consider raising our score, we value your feedback and will provide a response to it.
> > >
> > > Best regards,\
> > > Authors

---

### Official Review · Reviewer_Lbwj · 2026-03-15

**Soundness:** 3
**Presentation:** 3
**Significance:** 3
**Originality:** 3
**Overall Recommendation:** 2
**Confidence:** 4

**Summary:**

This paper points out that, in multimodal large language models, visual information is prone to being gradually lost as generation length increases during long-chain reasoning, making it difficult for these models to continuously benefit from longer reasoning processes in the same way as pure language models.
To address this issue, the authors propose VaLR, which inserts latent tokens aligned with image representations before each textual reasoning step, using them as dynamic “visual checkpoints” to continuously maintain visual grounding.
In terms of training, the paper adopts a two-stage scheme: it first performs conventional visual CoT fine-tuning, and then imposes a consistency constraint between the latent tokens inserted before each step and the features of an external visual encoder, thereby distilling fine-grained visual information into the reasoning process.
Experimental results show that VaLR significantly outperforms the baselines on multiple visual question answering and long-context reasoning tasks, improving the accuracy of Qwen2.5-VL on VSI-Bench from 33.0% to 52.9%, while also exhibiting a better test-time scaling behavior.

**Compliance With Llm Reviewing Policy:**

Affirmed.

**Key Questions For Authors:**

1. Since the objective of the representation alignment loss is to encourage the latent tokens to encode image-related visual information without selection, why not directly repeat the initial visual features into every round of the dialogue instead?

**Limitations:**

The article only includes an analysis of the positive application prospects of this method, lacking an analysis of limitations such as capability boundaries, resource and efficiency trade-offs

**Strengths And Weaknesses:**

# Strengths
1. The experimental section does not only compare absolute scores, but also analyzes the method from the perspectives of reasoning length, scalability, and the complementarity of different external visual encoders, which makes the experimental evaluation convincing.
2. The paper’s argument regarding test-time scaling is particularly strong. The authors show that the baselines degrade as reasoning length increases, whereas VaLR continues to improve, demonstrating the stable scalability and application potential of the proposed method.

# Weaknesses
1. The authors validate the method only on a single backbone, Qwen2.5-VL-7B, which is also somewhat outdated, and this is not sufficient to support the generality of the method across different MLLM backbones and model scales.
2. The authors claim that “visual information is gradually diluted during long-context generation,” but do not provide corresponding evidence, such as attention score distributions or representation utilization analyses, to support this claim. As a result, the motivation for the proposed method is not sufficiently convincing.
3. There are too few baselines included in the experiments to fully demonstrate the superiority of the proposed method, especially in comparison with traditional non-latent-reasoning methods. In addition, the performance shown in the tables still lags behind the current state-of-the-art results on these benchmarks to some extent.
4. Adding latent tokens is itself an operation that increases computation cost, yet the paper does not verify whether latent tokens are superior to language reasoning content under the same computation budget or context length.

---

> ### Author Rebuttal · Authors · 2026-03-31
>
> Dear Reviewer Lbwj,
>
> We sincerely appreciate your valuable comments and efforts. We do our best to clarify each of the concerns carefully as follows:
>
> ---
>
> **[W1] Generality of method in different backbones and model scales**
>
> To address generalizability, we trained VaLR to a broader set of models, including Qwen3-VL and InternVL-3.5  at multiple scales. As shown in Table below, VaLR consistently improves performance. For instance, Qwen3-VL-8B gains +7.1\%p and InternVL-3.5-14B gains +3.2\%p in VSI-Bench, which verifies strong generality of method.
>
> |  | VSI-Bench | BLINK | MMVP | MMStar | V\* | CVBench |
> | :---- | :---- | :---- | :---- | :---- | :---- | :---- |
> | Qwen2.5-VL-7B | 33.0 | 55.7 | 56.0 | 67.1 | 76.4 | 74.5 |
> | \+Ours | **52.9** | **64.7** | **60.3** | **72.3** | **86.9** | **87.6** |
> | Qwen3-VL-8B | 59.4 | 69.1 | 59.0 | 70.9 | 90.1 | 82.3 |
> | \+Ours | **66.5** | **70.8** | **67.3** | **77.4** | **92.1** | **91.7** |
> | InternVL-3.5-2B | 53.8 | 51.3 | 54.0 | 62.7 | 80.6 | 73.6 |
> | \+Ours | **60.5** | **62.4** | **61.0** | **70.2** | **85.8** | **87.7** |
> | InternVL-3.5-8B | 56.3 | 59.5 | 61.7 | 69.3 | 83.2 | 77.1 |
> | \+Ours | **62.1** | **66.9** | **68.7** | **76.8** | **90.1** | **88.2** |
> | InternVL-3.5-14B | 60.8 | 57.6 | 63.7 | 70.4 | 82.7 | 80.0 |
> | \+Ours | **64.0** | **69.7** | **70.3** | **77.7** | **91.6** | **90.9** |
>
> ---
>
> **[W2] Evidence of visual dilution in MLLMs**
>
> We have additionally analyzed visual dilution using the Prompt Dependency Measure (PDM) [1] following prior work [1,2,3], which quantifies dependence on visual input. As shown in the table below, both base models exhibit a clear PDM decline as generation progresses (e.g., Qwen2.5-VL drops by 63% from length 50 to 200), confirming progressive visual dilution. In contrast, VaLR maintains substantially higher and more stable PDM throughout generation (drops reduced to 15%), supporting our claim that VaLR effectively mitigates visual dilution.
>
> |  | 50 | 100 | 150 | 200 |
> | :---- | :---- | :---- | :---- | :---- |
> | Qwen2.5-VL | 0.38 | 0.28 | 0.20 | 0.14 |
> | \+Ours | **0.41** | **0.39** | **0.37** | **0.35** |
> | Qwen3-VL | 0.41 | 0.34 | 0.28 | 0.21 |
> | \+Ours | **0.46** | **0.44** | **0.43** | **0.42** |
>
> We will also include additional qualitative analysis based on attention score distributions in the revision.
>
> \
> [1] Favero, Alessandro, et al. "Multi-modal hallucination control by visual information grounding." CVPR 2024.
>
> [2] Zou, Xin, et al. "Look twice before you answer: Memory-space visual retracing for hallucination mitigation in multimodal large language models." arXiv (2024).
>
> [3] Jung, Mingi, et al. "Visual attention never fades: Selective progressive attention recalibration for detailed image captioning in multimodal large language models." arXiv (2025).
>
>
> ---
>
> **[W3] Limited baselines for non-latent-reasoning models**
>
> This is a good point. We extended our comparison to include a broader set of baselines, focusing on non-saturated benchmarks, i.e., VSI-Bench and BLINK. As shown in the table below, Qwen3-VL-8B with VaLR achieves SOTA performance on both benchmarks. Furthermore, Qwen2.5-VL-7B with VaLR outperforms 3D-specialized models such as VG-LLM and 3DRS which extensively exploits data modality-specific priors for improving the performance.
>
> |  | VSI-Bench | BLINK |
> | ----- | :---- | :---- |
> |  | *Open-source* |  |
> | Kimi-VL-A3B-2506 | 37.4 | 56.8 |
> | GLM-4.5V-106B-A12B | 41.4 | 65.3 |
> | InternVL-3.5-14B | 60.8 | 57.6 |
> | InternVL-3.5-38B | 66.3 | 60.9 |
> |  | *3D-LLM* |  |
> | VG-LLM-8B | 50.7 | \- |
> | 3DRS-7B | 45.9 | \- |
> |  | *Base model* |  |
> | InternVL-3.5-8B | 56.3 | 59.5 |
> | \+Ours | **62.1** | **66.9** |
> | Qwen2.5-VL-7B | 33.0 | 55.7 |
> | \+Ours | **52.9** | **64.7** |
> | Qwen3-VL-8B | 59.4 | 69.1 |
> | \+Ours | **66.5** | **70.8** |
>
> ---
>
> **[W4] Superiority of latent reasoning compared to language reasoning**
>
> We clarify that VaLR is more cost-efficient because it converges much faster. As shown in Figure 3, VaLR requires 10–20× fewer training steps than vanilla SFT to reach comparable performance.
> This suggests that latent reasoning helps the model acquire richer representations than language reasoning alone, leading to better performance with substantially lower data and computing cost.
>
> ---
>
> **[Q1] Lack of view selection mechanism**
>
> We clarify VaLR is designed to learn step-wise visual selection, not just preserve visual features. Through multi-view reasoning training, the model learns to dynamically attend to the representation of relevant views at each step. However, repeating the initial visual features as additional input tokens only provide static context and lack any adaptive selection mechanism.
>
> ---
>
> **[L1] Lack of limitation analysis**
>
> We will add an explicit limitations section discussing: (i) the reliance on model-based data annotation, and (ii) the additional training overhead (largely compensated by 10× faster convergence).

---

> > ### Author Rebuttal · Reviewer_Lbwj · 2026-04-07
> >
> > Thank authors for the rebuttal. The additional results are helpful, but they do not fully address my main concerns, so I will keep my original score.

---

> > > ### Author Response · Authors · 2026-04-08
> > >
> > > Dear Reviewer Lbwj,
> > >
> > > We sincerely thank you for acknowledging our rebuttal. We believe we have sufficiently addressed all of the raised concerns, and we respectfully request that you share any specific remaining questions or concerns so that we may have an opportunity to respond before the discussion period closes. Below is a brief summary of how each concern was addressed:
> > >
> > > ---
> > >
> > > **[W1] Generality across backbones and scales**: We added experiments on Qwen3-VL-8B and InternVL-3.5-{2, 8, 14}B, consistently demonstrating performance gains across diverse architectures and model scales. We note that **Reviewers 4kex, 2z6E, and sya1 agreed that this concern was satisfactorily addressed.**
> > >
> > > **[W2] Evidence of visual dilution**: We provided quantitative evidence via the Prompt Dependency Measure (PDM), showing that VaLR maintains stable visual grounding while base models degrade significantly over reasoning length. We note that **Reviewer 2z6E agreed that this concern was satisfactorily addressed.**
> > >
> > > **[W3] Limited baselines**: We extended comparisons to additional open-source and 3D-specialized models (e.g., GLM-4.5V, Kimi-VL, VG-LLM, 3DRS), and we believe these additional baselines sufficiently demonstrate the superiority of VaLR.
> > >
> > > **[W4] Comparison on same computation budget**: We provided clarification that VaLR converges 10–20× faster than language reasoning, and we believe this sufficiently demonstrates that VaLR is substantially more cost-efficient under comparable performance targets.
> > >
> > > **[Q1] Lack of visual selection mechanism**: We provided clarification that VaLR already performs dynamic, step-wise visual selection through multi-view reasoning training, which is fundamentally distinct from simply repeating static visual features, and we believe this sufficiently addresses the concern.
> > >
> > > ---
> > >
> > > \
> > > **As the raising concerns overlap with those of other reviewers, we gently ask you to revisit their comments and our responses.** If any concerns still remain despite the agreement from other reviewers, we would warmly welcome the opportunity to resolve the concerns through further discussion with all reviewers in the forum. We remain fully committed to addressing any concerns and kindly ask for your continued engagement.
> > >
> > >
> > > Sincerely,\
> > > Authors

---

### Decision · Program_Chairs · 2026-04-30

**Decision:**

Accept (regular)

**Comment:**

Overall, the paper tackles an important and underexplored issue in MLLMs, namely the degradation of visual information during long-chain reasoning, and proposes a simple latent-token-based alignment mechanism to mitigate this. The empirical results are generally good, especially on long-context reasoning tasks, and several reviewers find the method effective and practically promising. After rebuttal, most reviewers are positive and consider their concerns largely resolved, while a reviewer remains unconvinced, resulting in a somewhat mixed but overall slightly positive consensus.

The main strengths lie in the clarity of the problem setting and the simplicity of the proposed approach. The idea of inserting vision-aligned latent tokens to maintain visual grounding is intuitive and easy to integrate into existing frameworks. Empirical evaluation is relatively thorough, including analysis on reasoning length, scaling behavior, and multiple ablations, and the improvements on several benchmarks.

However, several weaknesses limit the strength of the contribution. First, concerns about generality and evaluation remain: although additional experiments are provided in the rebuttal, the original submission relies heavily on a limited set of backbones and benchmarks, and comparisons to stronger or more recent baselines are somewhat insufficient. Second, the core motivation (visual information dilution) was initially under-supported, and while the rebuttal provides additional evidence, more direct and comprehensive analysis should be provided in the main paper. Third, the method introduces additional computational overhead (latent tokens, alignment with external encoders), and although partial analysis is provided, the trade-offs with inference latency and efficiency are not fully characterized.

In summary, the paper presents a technically sound and empirically effective idea with clear potential, but with limitations in evaluation completeness, justification strength, and practical analysis. These issues place it at the borderline of acceptance.